# Design and Implementation of a Low-Power Biopotential Amplifier in 28 nm CMOS Technology with a Compact Die-Area of 2500 μm^2^ and an Ultra-High Input Impedance

**DOI:** 10.3390/s25072320

**Published:** 2025-04-05

**Authors:** Esmaeil Ranjbar Koleibi, William Lemaire, Konin Koua, Maher Benhouria, Reza Bostani, Mahziar Serri Mazandarani, Luis-Philip Gauthier, Marwan Besrour, Jérémy Ménard, Mahdi Majdoub, Benoit Gosselin, Sébastien Roy, Réjean Fontaine

**Affiliations:** 1Department of Electrical Engineering and Computer Engineering, Faculty of Engineering, University of Sherbrooke, Sherbrooke, QC J1K 2R1, Canada; lemw1601@usherbrooke.ca (W.L.); kouk2201@usherbrooke.ca (K.K.); maher.benhouria@usherbrooke.ca (M.B.); gaul2411@usherbrooke.ca (L.-P.G.); besm1401@usherbrooke.ca (M.B.); menj2310@usherbrooke.ca (J.M.); majm2404@usherbrooke.ca (M.M.); roys2121@usherbrooke.ca (S.R.); fonr2201@usherbrooke.ca (R.F.); 2Department of Electrical Engineering and Computer Engineering, Faculty of Science and Engineering, Laval University, Québec City, QC G1V 0A6, Canada; reza.bostani.1@ulaval.ca (R.B.); mahziar.serri-mazandarani.1@ulaval.ca (M.S.M.); benoit.gosselin@gel.ulaval.ca (B.G.)

**Keywords:** action potential neural signal, 28 nm CMOS technology, bio-potential amplifier, brain–machine interface, DC cancellation, low-noise, low-power, multi-channel, neural recording, high input impedance

## Abstract

Neural signal recording demands compact, low-power, high-performance amplifiers, to enable large-scale, multi-channel electrode arrays. This work presents a bioamplifier optimized for action potential detection, designed using TSMC 28 nm HPC CMOS technology. The amplifier integrates an active low-pass filter, eliminating bulky DC-blocking capacitors and significantly reducing the size and power consumption. It achieved a high input impedance of 105.5 GΩ, ensuring minimal signal attenuation. Simulation and measurement results demonstrated a mid-band gain of 58 dB, a −3 dB bandwidth of 7 kHz, and an input-referred noise of 11.1 μ$V_{\mathrm{rms}}$, corresponding to a noise efficiency factor (NEF) of 8.4. The design occupies a compact area of 2500 μ$m^{2}$, making it smaller than previous implementations for similar applications. Additionally, it operates with an ultra-low power consumption of 3.4 μW from a 1.2 V supply, yielding a power efficiency factor (PEF) of 85 and an area efficiency factor of 0.21. These features make the proposed amplifier well suited for multi-site in-skull neural recording systems, addressing critical constraints regarding miniaturization and power efficiency.

## 1. Introduction

Accurate readout interfaces are vital in medical devices for capturing physiological signals with high fidelity, enabling precise diagnosis and treatment [1,2]. In neurotechnology, neural recording systems are essential for monitoring brain activity, particularly in epilepsy research. The invasive recording techniques associated with the neurosurgical investigation and treatment of focal epilepsy provide an ideal opportunity to record neuronal firing patterns in the human brain. High-density neural recording is crucial for predicting epileptic attacks because it captures detailed information from many neurons across different brain regions [3,4]. This comprehensive data collection allows identifying subtle patterns and early warning signs that precede seizures, which might be missed with lower-density recordings. By analyzing a rich dataset, researchers and clinicians can develop more accurate and timely predictions, potentially improving intervention strategies and reducing the impact of seizures on patients’ lives.

Figure 1 illustrates the proposed block diagram of a multi-site neural recording system, comprising blocks such as acquisition, interface readout, and a central radio module. Amplification front-ends in neural recorders must fulfill several essential requirements for treating chronic diseases over prolonged periods. These encompass a substantial input impedance, minimal power usage to prevent tissue harm and optimize battery life [5,6,7,8], a compact area for system miniaturization, a low input-referred noise (IRN) for accurately detecting spikes in the presence of background noise, a substantial common-mode rejection ratio, and the elimination of DC voltage [9]. Furthermore, the demand for electrode arrays with increased density is growing. The interface necessitates low-noise amplifiers (LNAs), filters, data converters, and processing circuitry, such as neural spike detectors for in vivo data reduction, to fulfill high-density implementation.

Although current state-of-the-art designs offer acceptable noise performance, they face significant challenges in meeting the low power consumption and minimal silicon area requirements for high-density neural recording. This paper presents the design and evaluation of a compact, low-power integrated biopotential amplifier, specifically aimed at predicting epileptic seizures. This bioamplifier is designed for integration into a 49-channel neural recording Application-Specific Integrated Circuit (ASIC) currently under development [10,14]. The ASIC incorporates embedded spike compression and utilizes TSMC CMOS 28 nm technology. Notably, the ASIC integrates an active mechanism for suppressing low-frequency signals.

The bioamplifier demonstrates a bandpass frequency response and bypasses the need for passive AC coupling input networks. This design achieves one of the most compact sizes reported in the literature, while also striking an effective balance between low noise performance and power consumption.

The structure of this paper is as follows: Section 2 outlines the system requirements in detail. Section 3 reviews the latest DC-cancellation techniques developed. The proposed LNA circuit is explained in Section 4. Section 5 presents the methods used for simulating and measuring the LNA, with the design results discussed in Section 6. Section 7 provides a comparison with state-of-the-art designs. Finally, Section 8 offers the concluding remarks.

## 2. System Requirements

Signal Processing: The amplifier must proficiently process action potentials (AP)—low-frequency bioelectric signals primarily concentrated between 300 Hz and 10 kHz [15]. While the peak amplitude of APs is about 100 μV, it can surge up to several millivolts in abnormal scenarios involving superimposed activity from multiple neurons. The thermal and biological background noise picked up by the electrodes is about 10 μ$V_{\mathrm{rms}}$. Thus, considering the noise and full dynamic range of the signals, a recording front-end with 7–8 effective-number-of-bits (ENOBs) would suffice to digitize neural signals [10] faithfully.

Silicon Area Optimization: The amplifier should optimize the silicon area to accommodate the increased number of electrodes in a Multi-channel Electrode Array (MEA). As the electrode count increases, the recording interface pixel area must shrink to maintain a compact recording system size. This requirement stems from the need for dedicated electronic circuitry for each electrode, including signal amplification, filtering, and multiplexing. Surgical equipment constrains the size of the ASIC, as it will be implanted through a 5 mm diameter hole. Considering the package size, the maximum dimensions should be less than 1.4 mm × 1.4 mm for this 49-electrode neural recorder. Consequently, each channel should be limited to a 100μm×100μm area.

Power Consumption: The neural recorder must strictly adhere to a power consumption limited by battery-life constraints, the number of recording channels used, and thermal dissipation constraints of 0.8 $mWmm^{-2}$, to ensure that the temperature rise stays below 1 °C [16]. Thus, a 1.4 mm × 1.4 mm size should have a sub-milliwatt power consumption. Assuming a 50% power consumption, the neural recording front-end should be less than 10 μW per channel for a few tens of recording channels.

Noise Performance: Since the background noise picked up by the electrodes in the AP band is about 10 μ$V_{\mathrm{rms}}$, the IRN of the recording front-end in this band should be 4–8 μ$V_{\mathrm{rms}}$ [17].

Electrode Offset Tolerance (EOT): The amplifier must eliminate DC-offset voltages as high as 50 mV that may emerge at the recording system’s input due to the electrochemical nature of the electrode–tissue interface [15]. Eliminating this offset is crucial to avoid amplifier circuit saturation and ensure precise signal processing.

Input impedance: Low input impedances can draw excessive current from the electrodes, resulting in signal attenuation and loss of crucial neural information in the frequency range of interest (300 Hz–10 kHz). Therefore, the input impedance must exceed the Thevenin impedance of the electrode (approximately 5 nF) [15]. Additionally, the finite DC input impedance of the recording front-end can lead to DC currents at the electrode due to offset voltage, which may cause tissue damage over time. To prevent this, a DC input impedance Z_in_ greater than 1 GΩ is necessary to restrict the DC current to 50 pA for an offset voltage of 50 mV, ensuring safety and efficacy for most applications [18,19,20]. These requirements are summarized in Table 1.

## 3. Review of the State of the Art

A common approach to achieving low-frequency suppression in amplifier design is the use of capacitive feedback networks or AC coupling. This technique places large capacitors in series with the input electrode, to block DC offsets while maintaining signal integrity. Additionally, RC high-pass filtering is employed, utilizing the electrode–electrolyte capacitance in conjunction with a large resistor connected between the amplifier input and ground to reject low-frequency components effectively. This technique, notably used in the highly cited works of Prof. R. Harrison and Prof. Roman Genov, is known for its simple architectures, offering excellent noise performance with minimal power consumption, due to single-stage preamplifier designs [5,24,25,26,27,28]. However, the AC coupling architecture is prone to charging effects, occupies a substantial die area, and may require off-chip passive components to achieve high capacitance values, which have poor input impedance. RC filtering also demands large integrated resistors and complex biasing circuits. Implementing the resistor with a MOSFET in the triode region makes the biopotential amplifier highly sensitive to voltage bias and PVT variations. Based on our experience in analog circuit design, any bulk leakage currents in the pseudo-resistor can cause substantial DC bias shifts, which result in poor regulation of bias points in the front-end. This can severely affect the stability and performance of the circuit.

The chopper-stabilized amplifier, as another approach, is a widely used front-end topology for neural recording, as chopping effectively reduces the low-frequency flicker noise of the operational amplifier. A servo-loop is typically integrated to form a high-pass filter, attenuating the electrode offset at the output [29]. However, chopping and input capacitance limit Z_in_ to only a few MΩ, far below the required 1 GΩ. Whereas off-chip coupling capacitors can enhance Z_in_, they are impractical for a compact implant with an available surface of 100μm×100μm per channel.

One solution for making a compact design consists of using a traditional single-ended amplifier with a DC-servo loop (DSL) to regulate the dynamic DC offset [7]. Our work represents a new example implemented in 28 nm CMOS technology, strategically chosen to reduce the surface area and enable increased electrode density. This innovative implementation optimizes power consumption and maintains a satisfactory noise level, offering a promising solution for high-density neural recording systems. Moreover, this approach ensures enhanced input impedance and minimal signal attenuation compared to a capacitive feedback network. Finally, the input of the biopotential amplifier is isolated from the feedback output, so the system is robust to the impedance of recording electrodes.

## 4. Bioamplifier Circuit Design

By incorporating low-pass filtering within the feedback configuration, the method efficiently addresses the problem of DC offset voltage, resulting in substantial DC attenuation within the system transfer function [8,30]. The low-pass filter monitors the DC-offset voltage at the amplifier output. It performs a subtraction operation on the input, establishing a distinct high-pass characteristic at the system level [Figure 2a]. This approach requires extra active circuitry for high-pass filtering in the feedback loop, which increases the power consumption [31]. However, it has benefits like a lower input-referred noise, better DC rejection, and compact design. These features indicate that it could be a good option for biopotential amplifiers.

The proposed biopotential amplifier’s system diagram is depicted in Figure 2b, consisting of two single-ended operational transconductance amplifiers (OTAs). The amplifier’s output typically encounters a load capacitance C_L_ of around 150 fF. The feedforward amplifier OTA_1_ establishes the LNA’s low-frequency gain and low-pass cut-off frequency (*f*_1_), which consequently determines the pass-band gain.

To achieve a compact and low-noise design for the bioamplifier, the first stages of *OTA*_1_ and *OTA*_2_ must be optimized to minimize the number of transistors used. Among the various voltage amplifier architectures, *OTA*_1_ is responsible for providing the overall gain of the bioamplifier, which should exceed 40 dB [33,34]. Therefore, a simple two-stage Miller amplifier was selected for *OTA*_1_ to meet this requirement [Figure 3a]. In contrast, the amplifier used in the integrator feedback loop, *OTA*_2_, does not require high gain performance, as its primary function is to correct the output DC error. Consequently, a single-stage amplifier was chosen for *OTA*_2_ [Figure 3b].

The active Miller integrator in the feedback network comprises OTA_2_, a capacitor, and a resistor with a high value. The RC time constant (Ø) plays a role in controlling the −3 dB high-pass cut-off frequency of the bio-amplifier $(\tau =R_{eq}$·$C_{I})$. To achieve the desired high-pass cut-off frequency, both *R_eq_* and *C*_I_ need to have high values [14]. *R_eq_* was built using a non-tunable Quasi-Infinite Resistor (QIR) based on the Two Series Connected Outwardly with a Connected Gate MOS (TSOCGM) structure [35], as seen in Figure 3c. The QIR design guarantees a stronger resistance throughout the valuable voltage range thanks to its symmetric architecture. This characteristic makes the pseudo-resistor’s architecture less susceptible to having nonlinear effects on the LNAs’ performance. A comparison of the current–voltage characteristics of the proposed QIR with the conventional design can be found in [36]. This study makes use of 28 nm CMOS technology, where the relatively high threshold voltage of the transistors helps reduce leakage, effectively addressing concerns related to current leakage in the QIR implementation. This results in a steady output voltage with minimum variations, free from any voltage fluctuations associated with the QIR.

Equation (Equation 1) represents the total power of the IRN of the LNA proposed in [37,38], which consists of root mean square (RMS) values of the quadratic sum of the thermal noise component ($\bar{v_{th}}$) and flicker noise component ($\bar{v_{f}}$) for *OTA*_1_ and *OTA*_2_.

Meanwhile, this equation shows that, in the suggested circuit, connecting the output of OTA_2_ to the input of the LNA means that the IRN of the LNA is influenced by both the IRN of *OTA*_1_ and the output noise of *OTA*_2_.(1)$$\begin{array}{cc} v_{\mathrm{in},\mathrm{total},\mathrm{rms}}^{2} & =v_{\mathrm{in}1,\mathrm{th},\mathrm{rms}}^{2}+v_{\mathrm{in}1,f,\mathrm{rms}}^{2}+v_{\mathrm{out}2,\mathrm{th},\mathrm{rms}}^{2}+v_{\mathrm{out}2,f,\mathrm{rms}}^{2} \end{array}$$

Examining the differential amplifiers’ noise reveals that the primary noise originates from the input pair of the op-amp and current–mirror pair, neglecting the short-channel effect. Equation (Equation 2) presents a comprehensive breakdown of one OTA’s flicker and thermal noise contributions. To minimize the IRN of each OTA, it is essential to consider certain general factors. Firstly, biasing the input pair transistors in the subthreshold region proved effective in reducing the thermal noise, maximizing the *g_m_/I_D_* parameter. Secondly, incorporating large PMOS transistors (*M*_1_, *M*_2_) with significant *W* and *L* values in the input pair helped in minimizing flicker noise in both OTAs. These low-noise requirements result in considerable allocations of resources in terms of silicon surface area and power consumption for LNAs.

Adopting very-short-channel technology offers advantages in terms of area and power reduction. As transconductance directly correlates with transistor current, employing low-voltage, very-short-channel transistors presents a viable option for reducing the IRN simply by increasing the current. This eliminates the need for extra-large transistors (*WL*_1,2_, *WL*_3,4_) in the amplifier’s input stages. Bulky input transistors can lead to disadvantages such as increased input capacitance.(2)$$\begin{array}{c} v_{\mathrm{in},\mathrm{total},\mathrm{rms}}^{2}=\underset{\mathrm{Thermal Noise}}{\underset{︸}{8kT\left(\frac{2}{3g_{m1}}+\frac{2g_{m3}}{3g_{m1}^{2}}\right)}}\;+\underset{\mathrm{Flicker Noise}}{\underset{︸}{\frac{2K_{P}}{C_{ox}{\left(WL\right)}_{1}f}+\frac{2K_{N}}{C_{ox}{\left(WL\right)}_{3}f}\frac{g_{m3}^{2}}{g_{m1}^{2}}}}\; \end{array}$$

## 5. Methods

Characterization of the design involved conducting simulations at the post-layout stage, which included extracting both the parasitic capacitance and resistance. Subsequently, the results obtained from benchtop testing of the fabricated LNA will be reported and discussed.

### 5.1. Frequency Bandwidth and Mid-Band Gain

Simulating the output amplitude in response to a 1 V input AC signal (*V_in_*), the AC simulator generated a frequency-dependent curve that resembled the amplitude characteristics depicted in the output Bode diagram shown in Figure 2a. The maximum amplitude of this curve was the mid-band differential gain (*A_v_*), and the frequency range between −3 dB high-pass (*f_HP-3dB_*) and low-pass cut-off frequencies (*f_LP-3dB_*) was the bandwidth (BW) of the LNA.

Regarding the experimental BW and mid-band gain measurements, Figure 4a depicts the test bench used to measure these AC parameters. An Agilent 35670A dynamic signal analyzer (DSA), with high measurement resolution and input level noise less than −130 dB$V_{\mathrm{rms}}$/$\sqrt{\mathrm{Hz}}$, was utilized to characterize the LNA. A frequency sweep used the [Sweep Sine] mode to obtain the AC gain curve within the desired bandwidth.

Utilizing resistors (*R*_1_, *R*_2_) to create an attenuation factor offered several advantages. Firstly, since most signal generators cannot generate signals below approximately ~100 mV, an additional attenuation network became necessary to produce microvolt-level signals for testing purposes. Secondly, the Agilent 35670A can introduce noise that contributes to the overall output noise of the LNA. To mitigate this, an attenuation block can be added before the LNA, effectively reducing the noise from the AC source. Consequently, the input AC signal from the signal generator can increase the amplitude, while decreasing noise.

However, these resistors introduce another source of thermal noise, as follows:(3)$$v_{n}^{2}=4kT\left(R_{1}\|R_{2}\right)\;\cdot \;BW,$$
where *BW* is the noise bandwidth of the device under test. Keeping *R_1_* <100 Ω is typically sufficient to reduce this noise source to insignificance. Given that an attenuation factor of 100 is required with a source voltage of 100 mV, *R_2_* = 10 kΩ was chosen. The AC analysis can be performed by adjusting over ten sampling points per frequency decade to achieve a satisfactory resolution.

### 5.2. CMRR

We stick with the traditional way of defining CMRR for op-amps, which is described by(4)$$\mathrm{CMRR}\left(\mathrm{dB}\right)=20\log\left(\frac{A_{v}}{\left|A_{CM}\right|}\right),$$
where *A_v_* represents the differential gain, and *A_CM_* stands for the maximum common-mode gain of the amplifier in BW.

To practically measure *A_CM_*, the same procedure was followed as to measure midband gain, but the input terminals of the LNA (*V_in_* and *V_ref_*) were tied together to have the same input signal *V_CM_* [39] (Figure 4b). This was used to generate a curve of the *A_CM_*’s amplitude versus frequency of an *A_CM_* from *f_HP-3dB_* to *f_LP-3dB_*, to find the maximum value. Our fabricated LNA includes a low-pass RC filter within the ASIC, to mitigate noise from the integrated reference voltage source.

### 5.3. Power Supply Rejection Ratio (PSRR)

This involves sweeping the signal frequency across the amplifier’s bandwidth. The PSRR is calculated as follows:(5)$$\mathrm{PSRR}=20\log\frac{A_{v}}{A_{dd}\left(V_{in}=0\right)},$$
*A_dd_* is the gain influenced by power-supply ripple when the differential input is at zero [40].

To introduce a power supply ripple, we inserted an AC source separately into the power supply rails +*V_dd_* and −*V_SS_*. This action enabled the calculation of PSRR+ and PSRR− without applying an AC signal to the amplifier’s differential input. Figure 4c shows one of the possible experimental test benches to measure the PSRR. It was necessary to verify the DC and AC levels of the power signal and signal generator delivered to the LNA. The curve of the *A_dd_*’s amplitude per frequency was expected, including a bandwidth from *f_HP-3dB_* to *f_LP-3dB_*, to find the maximum of *A_dd_*.

### 5.4. Input-Referred Noise

Simulation of the IRN in RMS is achievable through Cadence Virtuoso noise analysis tools (Noise Summary). This process involves probing the LNA’s input and output nodes under a specified DC bias to extract the RMS noise within the system’s bandwidth. To determine the complete RMS noise, incorporating both thermal and flicker noise, the RMS output noise must be divided by the LNA’s bandpass gain.

IRN was measured as shown in Figure 4d. An RC low-pass filter canceled any noise from the bias circuits with R = 10 kΩ, C = 10 μF. Then, to achieve a precise noise measurement of the fabricated LNA, in terms of *V_rms_*, both digital and analog measurement instruments were applied in parallel. An Agilent MSO-X 2024A, which features mathematical functions and real-time calculation capabilities, was utilized to capture the LNA’s Fast Fourier Transform (FFT) when only the DC bias was present on the inputs. An oscilloscope provided the RMS voltage for each FFT sample. Then, the FFT curve was converted to a Power Spectrum Density (PSD) according to (Equation 6).(6)$$\mathrm{PSD}\left(X_{i}\right)=\frac{{\left(X_{i}\right)}^{2}}{\Delta f},$$
where *X_i_* is an FFT sample in RMS scale, and Δf is the sampling frequency resolution.

The RMS value of the IRN was the integrated area under this curve and divided by mid-band gain over the desired bandwidth. Numerical integration was performed simply by adding the FFT samples between the frequencies *f_HP-3dB_* and *f_LP-3dB_*, inclusively, according to (Equation 7).(7)$$V_{\mathrm{in},\mathrm{rms}}=\frac{\sqrt{\sum_{i=f_{\mathrm{HP}-3\mathrm{dB}}}^{i=f_{\mathrm{LP}-3\mathrm{dB}}}\mathrm{PSD}\left(X_{i}\right)}}{A_{v}}$$

The RMS voltmeter URE3 is an analog measurement instrument with less than 10 μ$V_{\mathrm{rms}}$ input noise. Thus, this provided testing means equivalent to a digital approach (Figure 4d). Choosing an AC-RMS mode RMS meter gave the RMS value of the output noise of the LNA in a selected BW. Then, the measured value was divided by *A_v_* to obtain the RMS of the LNA’s IRN.

### 5.5. Input Impedance

To characterize the input impedance of the LNA, a voltage drop was measured across a 10 MΩ resistor (*R_3_*) connected in series with the LNA input [41]. This voltage was sensed using a difference amplifier (OPA227) and the DSA, as illustrated in Figure 4e. The corresponding input impedance was calculated as(8)$$Z_{\mathrm{in}}=R_{3}\left(\frac{R_{5}}{R_{4}\times G_{\mathrm{DSA}}}-1\right)$$
where $G_{\mathrm{DSA}}$$=\frac{V_{\mathrm{Input}}}{V_{\mathrm{Source}}}$ is the gain measured by the DSA during the experiment.

### 5.6. Electrode Offset Tolerance

The methodology for assessing the proposed LNA’s EOT involved measuring the maximum DC variation range of the output. This measurement consisted of applying a variable input offset to the LNA, ranging from GND to *V_DD_*. Here, the observed change in the output DC level for each change in the input DC level within this range was introduced as the DC cancellation gain.

### 5.7. Linearity

Numerous articles have relied on total harmonic distortion (THD) to explain linearity. However, in this study, our central focus for spike-recording applications is the gain reduction caused by interference like electromagnetic interference or low-frequency local field potentials, leading to varying gains over time [42]. Hence, we proposed that assessing the −1 dB gain compression point (roughly 89% of the voltage gain) was more practical for characterizing these amplifiers than using THD.

The test setup for measuring the −1 dB gain compression was the same as that for measuring the mid-band gain [Figure 4a]. The LNA was connected to a signal generator, spectrum analyzer, or power meter. Then, the signal generator was set to output a sine-wave signal at a representative frequency. By gradually increasing the input signal voltage, we identified where the gain dropped by 1 dB from the linear gain [43]. This was the −1 dB gain compression point. The linearity performance could be evaluated by comparing the input voltage for dB gain compression.

### 5.8. Efficiency

To evaluate our amplifier’s noise and power capabilities against others, we employed the noise efficiency factor (NEF) [44] described by(9)$$\mathrm{NEF}=v_{in,rms}\sqrt{\frac{2I_{tot}}{\pi \;\cdot \;U_{t}\;\cdot \;4kT\;\cdot \;\mathrm{BW}}},$$
where *I_tot_* is the total amplifier supply current that was measurable by a Model 6487 Picoammeter/Voltage Source. This instrument has a measurement resolution of 10 fA. *U_t_* is the thermal voltage and *k*, the Boltzmann constant is defined as $1.38\times 10^{-23}$ J·$K^{-1}$.

When comparing two circuits operating at the same supply voltage, the NEF is a suitable measure to assess the power-to-noise balance. However, if two amplifiers possess identical total current and noise yet operate at different V_DD_ values, their NEF might be the same, even though their power usage differs. This indicates that the NEF alone is not adequate for evaluating power efficiency. To address this concern, a more direct evaluation of overall power consumption can be achieved using the Power Efficiency Factor (PEF) metric [45]. The PEF normalizes the product of noise power and total power, resulting in $\mathrm{PEF}={\mathrm{NEF}}^{2}\;\cdot \;V_{DD}$.

Although the PEF was adequate to compare the performance of the amplifiers in terms of power consumption and with the same die area, it assesses the merits of amplifiers when the silicon occupation surface is an essential parameter. In this work, the priority was designing a very small-sized LNA with a very dense neural recorder. Therefore, a different metric, designated Area Efficiency Factor (AEF), was proposed(10)$$\mathrm{AEF}=\mathrm{PEF}\times \frac{A}{10^{-6}},$$
where *A* is the total silicon area of the LNA, including its capacitors.

## 6. Results

A dedicated test bench hardware and setup were developed for the fabricated ASIC, integrating communication modules and acquisition units to evaluate the ASIC as a wireless multi-site neural recorder. This test bench is discussed in [46].

The test PCB, illustrated in Figure 5a, was specifically designed to assess the LNAs’ performance while mitigating noise interference from the ASIC’s digital circuitry. A battery-powered LDO regulator ensured a stable, low-noise supply voltage for the chip. Additionally, multiple SMA connectors facilitated seamless interfacing between the amplifier, signal sources, and measurement equipment.

Figure 5b presents the readout interface, fabricated using TSMC’s 28 nm process, while Figure 5c illustrates the biopotential amplifier layout, which operates at a 1.2 V supply voltage and occupies a compact 2500 μ$m^{2}$ area. To evaluate the stability of the ASIC, key parameters such as output noise, power consumption, and frequency response were monitored over 24 h across five ASICs at room temperature.

### 6.1. Frequency Bandwidth and Mid-Band Gain

The simulation was performed at *f_HP-3dB_* = 600 Hz and *f_LP-3dB_* = 7 kHz for the Typical–Typical (TT) process corner. Figure 6 shows the statistical distribution of mid-band gain obtained using 800 Monte Carlo simulations. The results include the local and global mismatches, depending on the process corners. When $V_{DD}$ changed by 10%, the average gain varied from 55.8 to 58.2 dB. Moreover, the LNA mid-band gain depending on temperature was investigated using circuit simulation (Figure 7a).

The measured AC response (Figure 8) shows that the mid-band gain was 57 dB. The −3 dB high-pass corners occurred at approximately 150 Hz, and the −3 dB low-pass corner was 7.1 kHz. The gain variation measured across five LNAs on different chips was less than 1 dB.

The difference between the simulated and measured *f_HP-3dB_* was notable, as illustrated in Figure 2a. This discrepancy was likely due to the considerable impact of variations in $R_{eq}$, as well as the gain of OTA_2_ and the capacitance charge from the measurement probe, which can shift *f_HP-3dB_* to a lower frequency. However, this cutoff still effectively attenuated the majority of the LFP components, which predominantly occupy frequencies below 100 Hz. Any residual LFP content that might have been present in the recorded signal was further mitigated by the spike detection and processing stages.

### 6.2. CMRR

Post-layout simulations were performed for all amplifiers, yielding a simulated CMRR of 78 dB for *CMRR_OTA1_* and 70 dB for *CMRR_OTA2_*. However, the measured CMRR for the LNA was notably lower at 55 dB in the bandwidth.

### 6.3. PSRR

With a simulated voltage gain of $A_{v}=55$ dB and supply gains, $A_{dd+}=-5$ dB and $A_{dd-}=7$ dB, from positive and negative supply, respectively, the simulated PSRR was calculated as follows: PSRR = min {PSRR_−_ = 60 dB, PSRR_+_ = 48 dB} = 48 dB. Figure 9 shows the measured PSRR of the fabricated LNA. According to this curve and (Equation 5), the worst measured PSRR of the LNA in the bandwidth (150 Hz to 7.1 kHz) was 52 dB.

### 6.4. Noise Measurement

In this work, the IRN from the post-layout simulation was 8.4 μV_rms_ when the mid-band gain was 57 dB for a 600 Hz to 7 kHz bandwidth. Figure 7b illustrates the simulated IRN across a temperature range from −20 °C to 80 °C.

In Figure 10, the Power Spectrum Density (PSD) of the noise in the LNA output is depicted, calculated from the FFT analysis of the measured noise. The RMS value of the output in the bandwidth was 13.4 mV_rms_, which gave a total IRN of 15.8 μV_rms_. Table 2 shows the contribution of the different noise sources besides the LNA, such as the input’s electrostatic diode (ESD) and 600 mV reference voltage. In this way, the estimated IRN of the LNA was 11.1 ± 2.7 μV_rms_.

### 6.5. Input Impedance

At DC frequencies, particularly below 1 Hz, a typical frequency where the biopotential front-end should reject dynamic DC offsets, the AFE exhibited an input impedance of 105.5 GΩ. As the frequency increased, this impedance dropped to 1.1 GΩ at 150 Hz, 195 MΩ at 1 kHz, 29.5 MΩ at 7 kHz, and 19.3 MΩ at 10 kHz, where the biomedical AFE is characterized. Figure 11 shows the measured input impedance of the proposed AFE versus the simulated input impedance across the frequency range of LNA operation. The comparison between the measured and simulation input impedance shows that the PCB routing, PAD, and parasitic ESD configured the parallel connection with the input impedance of the AFE, and has potential to reduce the total input impedance. Figure 7c presents the simulated input impedance at −20 °C, 27 °C, and 80 °C. While the DC input impedance decreased significantly at higher temperatures, it remained nearly constant within the AP bandwidth.

### 6.6. Electrode Offset Tolerance

In Figure 12, the measured output dynamic DC offset is illustrated concerning DC input voltage bias ranging from 0 to 1.2 V. The graph indicates that the LNA avoided saturation, even with a variation in DC input from 0 to 910 mV. As depicted in the diagram in Figure 12, the DC cancellation gain of the LNA was determined to be −30 dB.

### 6.7. Linearity

The measured result indicates that the −1 dB gain compression point was observed at an input level of 1.4 mV.

### 6.8. Efficiency

A dedicated supply pin was utilized to assess the LNA current, providing power to the test LNA alongside its comparator. Initially, the current of this LNA–comparator combination was measured for five ASICs, yielding an average of 4.9 μA. Subsequently, through simulation of the comparator in all five process corners, the average comparator consumption was determined to be 2.1 μA. By deducting the comparator’s current from the overall measured current, the LNA’s current consumption could be calculated at 2.8 μA when supplied with 1.2 V, which agrees reasonably well with the simulation results.

Based on the measured BW of 7 kHz and the measured IRN level of 11.1 μV_rms_, the NEF of the LNA was determined to be 8.4. With a supply voltage of 1.2 V, the PEF was calculated to be 85. Then, assigning a layout area of 2500 μ$m^{2}$ to the LNA, an Area Efficiency Factor (AEF) of 0.21 was obtained.

Besides the TT process corner, four other process corners, namely Fast–Fast (FF), Fast–Slow (FS), Slow–Fast (SF), and Slow–Slow (SS), are reported in Table 3 as simulations results.

## 7. Discussion and Comparative Examination

### 7.1. Neural Signal Recording Using 28 nm CMOS Technology

Designing low-noise devices using a circuit with very-short-channel technologies is quite challenging. This is mainly because the way transconductance behaves in 28 nm CMOS technology is not straightforward—it does not just increase with more drain current or transistor area (*WL*), which goes against (Equation 2). The suggested architecture, where a feedback loop is directly connected, has trouble with nonoptimal PSRRs and CMRRs. In addition, the noise from OTA_2_’s output becomes the input for OTA_1_. As a result, this noise keeps getting amplified by the LNA. Furthermore, due to the high impedance of the electrode on one input of feedforward amplifier (OTA_2_), the LNA exhibits an inadequate CMRR. A fully differential architecture could be employed for the proposed LNA to enhance both the PSRR and CMRR. However, this approach would result in a twofold or more increase in power consumption, die area, and IRN.

### 7.2. Comparative Assessment with Previous Studies

Table 4 compares this study and previous works on DC-coupled LNAs that utilized technologies featuring longer channel lengths. Adopting a single-ended architecture may seem unconventional, but it was a deliberate choice driven by practicality, power efficiency, and silicon area considerations. Figure 13a compares the area and power consumption between previous state-of-the-art designs. Single-ended LNAs exhibit greater sensitivity to common noises, such as CMRR and PSRR. However, in the context of our research, a fully differential architecture would lead to a bulky implementation. This choice becomes critical when emphasizing the necessity for an ultra-compact design accommodating 49 channels within a limited 1 mm^2^ space.

The proposed neural amplifier achieved an input-referred noise of 11.1 μVrms, which, while slightly higher than some referenced designs (Table 4), remains sufficient for detecting spikes with amplitudes exceeding 50 μV. As demonstrated by [47], extracellular spike amplitudes typically range from ∼ 50 μV to 500 μV, with most spikes from nearby neurons (>50 μm) surpassing 70 μV. In high-density recording systems (e.g., Utah arrays, Neuropixels), where electrodes are positioned close to active neural populations, the majority of detectable spikes are expected to exhibit amplitudes well above this threshold. Thus, the amplifier’s noise performance remains compatible with reliable spike detection in such configurations.

Meanwhile, the proposed design achieves a considerable reduction in silicon area compared to prior designs. For instance, the proposed architecture reduced the chip area by at least fivefold relative to [45], enabling a 2.2× decrease in inter-electrode spacing. This miniaturization facilitates a higher electrode density, which offers two critical benefits:Improved recording spatial resolution: Closer electrode spacing enhances the ability to resolve individual neural units and reduces signal cross-talk.Enhanced signal-to-noise ratio (SNR): By positioning electrodes nearer to neural sources, the amplitude of recorded spikes increases, improving detectability and selectivity.

Future work will involve in vivo validation to assess the amplifier’s performance in biological settings, particularly its ability to capture high-fidelity neural signals in densely packed electrode arrays. These experiments will further clarify the trade-offs between noise, area efficiency, and spike detection efficacy in next-generation neural recording systems. Figure 13b alongside Table 4 further provides well-estimated NEF and PEF metrics and the area efficiency factor. These additional insights can contribute to a more thorough understanding of the trade-offs made in our design choice.

**Table 4 sensors-25-02320-t004:** Comparison with prior state of the arts.

Reference	This Work	VLSI’23 [21]	TCAS-II’21 [48]	JSSC’17 [8]	JSSC’11 [45]	TCAS-II’20 [49]
**Tech.**	28 nm	180 nm	180 nm	40 nm	65 nm	180 nm
**V_DD_ (V)**	1.2	1.8	0.8	1.2	0.5	1.2
**Power (μW)**	3.4	13.9	0.52	2.8	5.13	3
**Area (mm^2^)**	0.0025	0.085	0.24	0.069	0.013	0.2
**Input Impedance**	105.5 GΩ @DC	64 MΩ @60 Hz	-	1.6 GΩ @DC	-	-
**EOT (mV)**	910	50	-	-	100	±300
**Gain (dB)**	57	40	40	25.7	-	40
**BW (Hz)**	150–7.1k	1–100	800	0.12–5k	300–10k	0.5–200
**Noise (upmuV_rms_)**	11.1	0.59	1.1	5.6	4.9	0.67~1.49
**PSRR (dB)**	52	-	75	76	50	-
**CMRR (dB)**	55 @BW	106 @50 Hz	104	78	75	-
**NEF**	8.4	6.4	2.1	4.4	6	2.88~6.41
**PEF**	85	73.7	1.2	23.2	18	10~49.3
**AEF**	0.21	6.3	0.29	1.6	0.23	1.99~9.86

**Figure 13 sensors-25-02320-f013:**
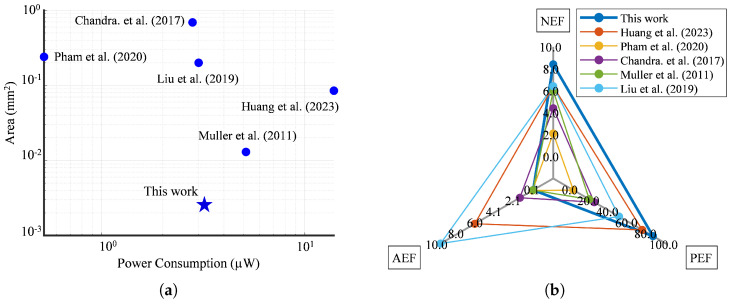
Comparison with the state of the art presented in Table 4 [15,21,45,48,49]: (**a**) Scatter plot of silicon area occupation versus power consumption. (**b**) Radar chart comparing NEF, PEF, and AEF.

## 8. Conclusions

In conclusion, this paper introduced a DC-coupled biopotential amplifier that replaces traditional DC-blocking capacitors with analog feedback. This feedback mechanism effectively senses and cancels the DC offset at the input by monitoring the output of the amplifier. The measured input impedance of the system was confirmed to be sufficiently high, ensuring compatibility with high-impedance neural electrodes. This guarantees minimal signal attenuation, preserves neural signal integrity, and maintains the system’s ability to operate effectively in neural recording applications. The implementation of this approach, combined with the utilization of a 28 nm CMOS node, resulted in a significant reduction in the required area. Consequently, a recording channel could be seamlessly integrated into a compact 100 μm × 100 μm surface, facilitating the development of a 49-channel neural recording ASIC designed for an epilepsy prediction device. Moving forward, our future work will focus on developing a comprehensive recording system on a single chip, aiming to achieve an even denser arrangement of channels and ultimately enhancing the spatial resolution of the recorded neural data.

## Figures and Tables

**Figure 1 sensors-25-02320-f001:**
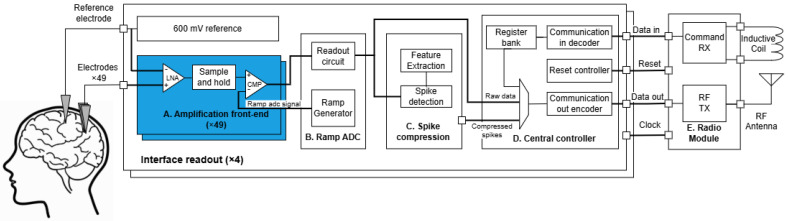
Block diagram of the neural recorder. Each of the 49 pixels at each site contains a front-end circuit (**A**) with a biopotential amplifier, a sample-and-hold circuit, and a comparator. A ramp ADC (**B**) distributes a sawtooth waveform to all pixels, triggering the comparator when it intersects the amplified electrode signal. The readout circuit captures this transition, outputting the digitized value and electrode address to the spike detection and compression circuit (**C**). A central controller (**D**) oversees communication and configuration via a register bank [10]. The radio module (**E**) features an ultra-wideband uplink transmitter, a narrowband downlink receiver, and an inductive wireless power transmitter sharing a common coil [11,12,13].

**Figure 2 sensors-25-02320-f002:**
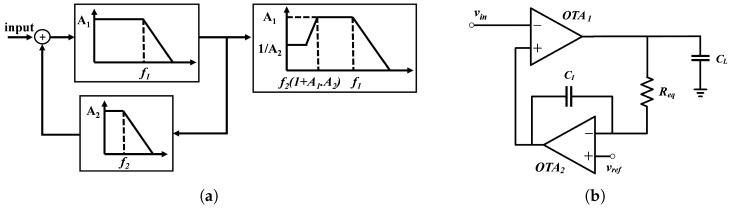
DC cancellation architecture. (**a**) Frequency response of high-pass filter implementation using low-pass filter in feedback [32]. (**b**) Proposed LNA systemic schematic with high-pass filter implemented through low-pass filter feedback.

**Figure 3 sensors-25-02320-f003:**
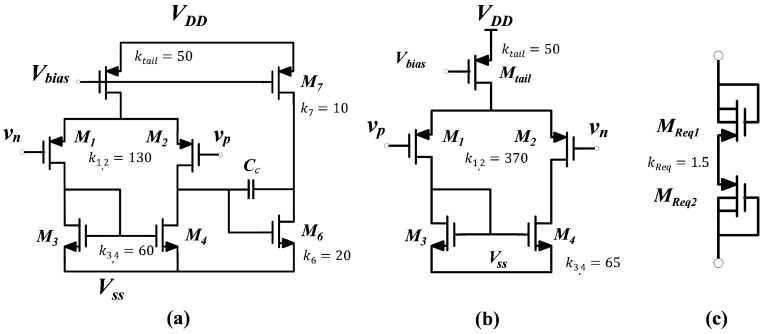
Circuit diagrams of (**a**) $OTA_{1}$, (**b**) $OTA_{2}$, (**c**) $R_{eq}$.

**Figure 4 sensors-25-02320-f004:**
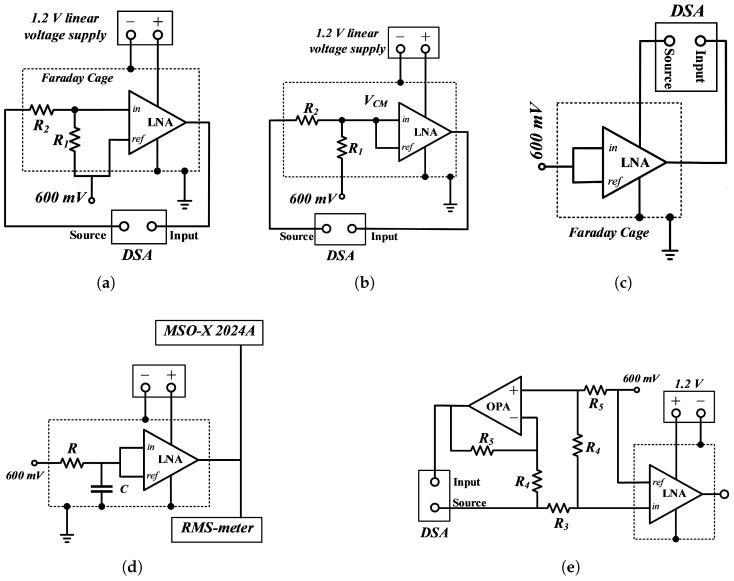
Experimental test setups. (**a**) The schematic for measuring the AC gain of the LNA. (**b**) Test bench for measuring CMRR. (**c**) PSRR setup, obtaining the gain of the LNA when the AC input was from the supply ($A_{dd}$). (**d**) Experimental setup with input low-pass filter to measure the output noise of the fabricated LNA. (**e**) Input impedance measuring.

**Figure 5 sensors-25-02320-f005:**
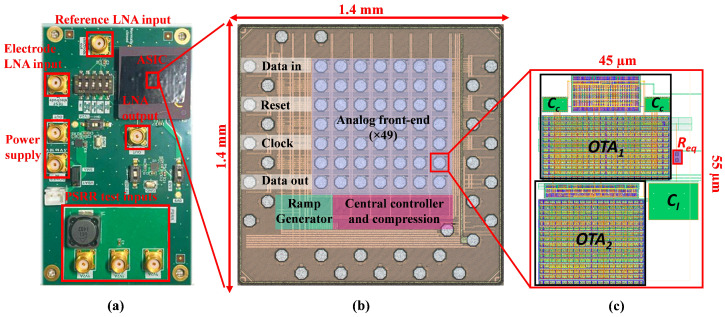
(**a**) Test PCB for analog front-end measurement. (**b**) Micrograph of the ASIC featuring 49 recording channels. (**c**) The layout of the proposed LNA was designed using 28 nm CMOS technology.

**Figure 6 sensors-25-02320-f006:**
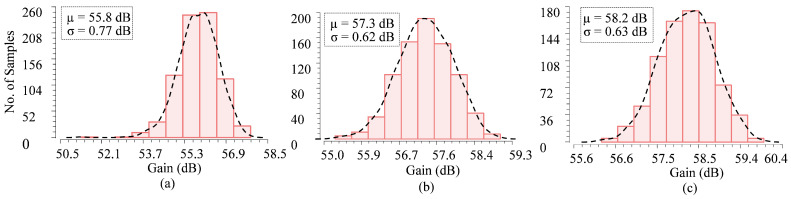
Statistical distribution of the mid-band gain of the LNA. (**a**) *V_DD_* = 1.2 V, (**b**) *V_DD_* = 1.1 V, (**c**) *V_DD_* = 1.3 V.

**Figure 7 sensors-25-02320-f007:**
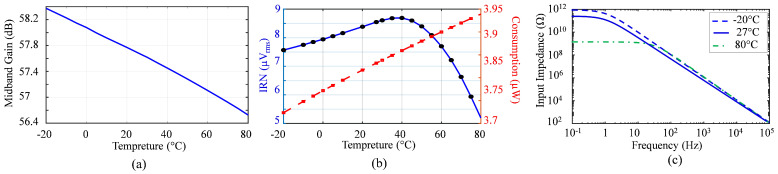
Temperature variation simulation of the LNA, showing (**a**) mid-band gain, (**b**) input-referred noise (blue solid) and power consumption (red dash), and (**c**) input impedance across different temperatures.

**Figure 8 sensors-25-02320-f008:**
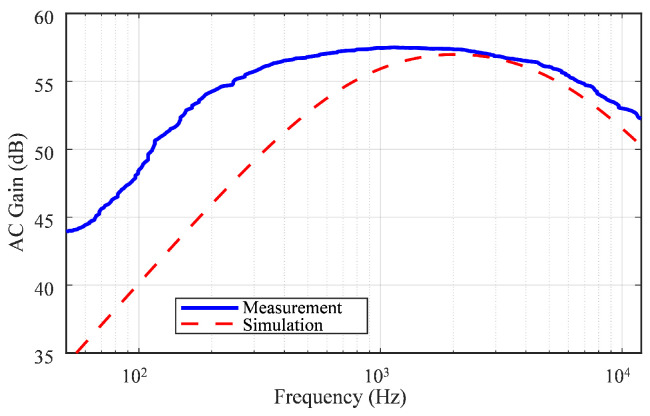
Comparison of measured and simulated gain magnitude responses of the LNA.

**Figure 9 sensors-25-02320-f009:**
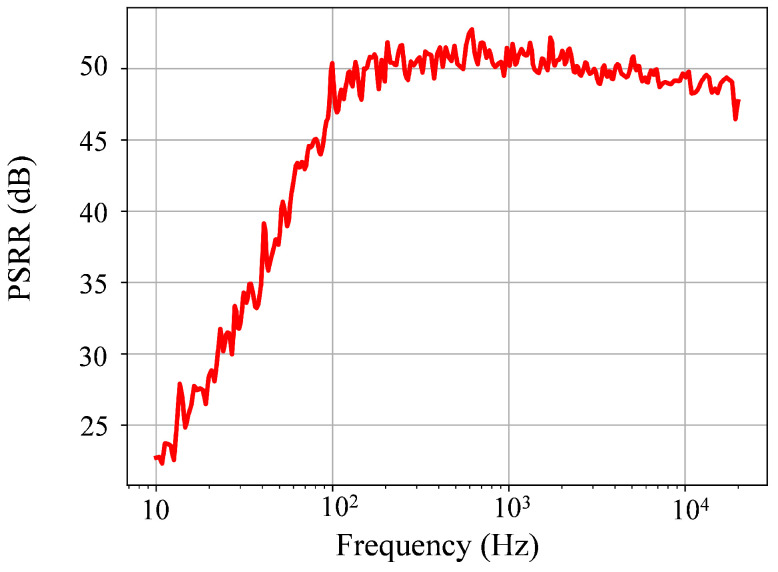
Measured PSRR of the LNA.

**Figure 10 sensors-25-02320-f010:**
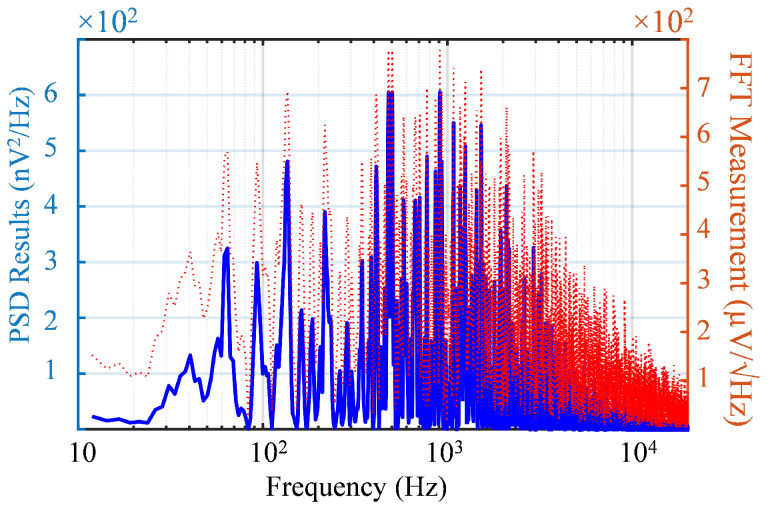
Calculated PSD from the measured FFT output noise. The solid blue line represents the calculated PSD, while the red dashed line corresponds to the measured FFT output.

**Figure 11 sensors-25-02320-f011:**
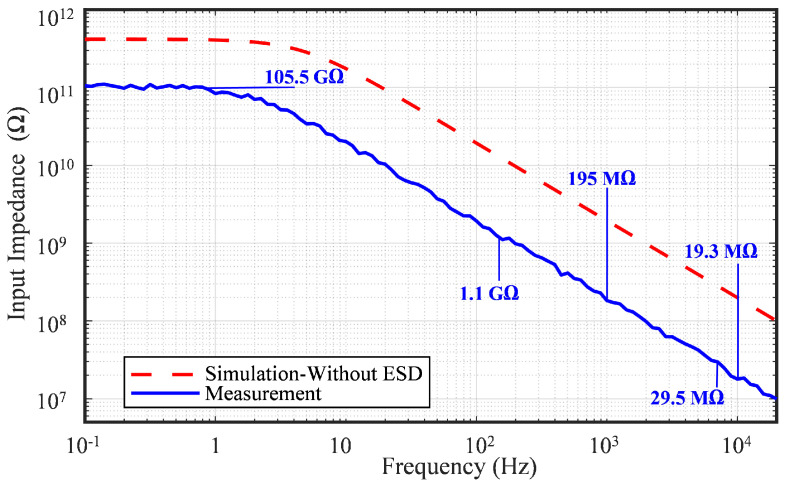
Measured input impedance derived from the voltage drop across a 10 MΩ resistor, compared to the simulated impedance values without ESD protection using the DSA.

**Figure 12 sensors-25-02320-f012:**
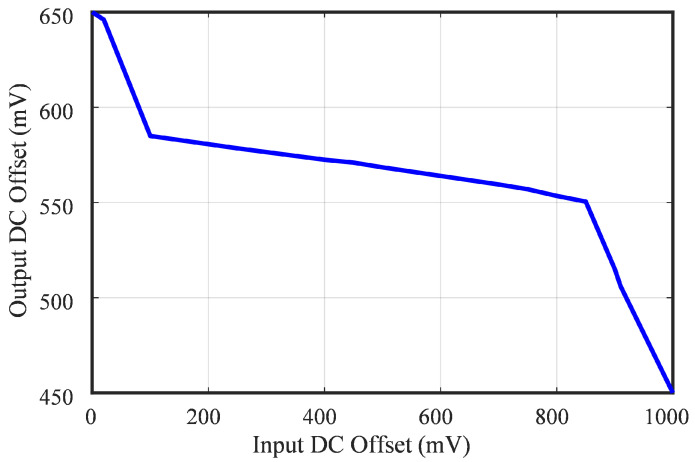
Measured dynamic DC offset and DC cancellation performance of the LNA.

**Table 1 sensors-25-02320-t001:** System requirements and summary of the state of the art.

Parameter	[21]	[22]	[23]	Required
**Power/ch (μW)**	13.9	2.3	1.19	**<10**
**BW (Hz)**	1–100	300	0.1–100	**300–10k**
**EOT (mV)**	50	50	400	**>50**
**IRN (μV_rms_)**	0.59	7	0.91	**4–8**
**Area/ch (mm^2^)**	0.085	0.025	0.41	**<0.01**
**DC Z_in_**	64 M @60 Hz	28 M @100 Hz	469 M @50 Hz	**>1 G**

**Table 2 sensors-25-02320-t002:** AP noise partitioning test of LNA.

Noise Source	IRN (μV_rms_)
**IRN_OTA1_ ***	6.4
**IRN_OTA2_ ***	5.5
**ESD ***	3.3
**V_ref_**	10 ± 3
**Total**	15.8

* Simulation.

**Table 3 sensors-25-02320-t003:** Comparison table for different process corner simulations.

Process	TT	FF	FS	SF	SS
**IRN (μV_rms_)**	8.4	8.5	8.8	8	8.5
**Gain (dB)**	57	55.7	56.7	57.1	58.1
**Output noise** **(mV_rms_)**	6	5.2	5.9	5.8	6.9
**Power (μW)**	3.55	3.71	2.36	3.61	2.34
**BW (kHz)**	0.6–7	1.45–12.2	0.37–5.2	1.1–8.4	0.33–3.9

## Data Availability

No new data were created or analyzed in this study.

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
