# Peer review of "Design and Implementation of a Low-Power Biopotential Amplifier in 28 nm CMOS Technology with a Compact Die-Area of 2500 μm2 and an Ultra-High Input Impedance"

_sensors, 2025, doi:10.3390/s25072320_

Round 1
Reviewer 1 Report
Comments and Suggestions for Authors
This paper presents the design and implementation of a biopotential amplifier for AP signal recording. The proposed circuit achieves compact area and ultra-high input impedance, which are widely recognized as key indicators for the front end of neural signal acquisition. Although the circuit technology used in this paper is already known, addressing and overcoming limitations (e.g., the 49-electrode arrays, leakage properties of the 28nm node process itself, etc.) makes this paper's exploration of multi-channel applications in deep submicron technology very valuable. This is also the most positive aspect of this article. The article also provides the explanation of the experimental testing of the implemented amplifier. However, there are some questions for the authors to improve the quality of the paper:
- In line 88, the authors mention “Electrode Offset Tolerance (EOT) and input impedance.” However, the following section only characterizes the input impedance, and I do not see an explanation of the EOT.
- In lines 99-102, the authors write “First…, Second…, Third….” The first and third points may describe the same amplifier structure, as exemplified by the work of Professor Harrison, which the authors subsequently reference. If the authors intend to describe three different structural features, either the first or third point needs to be explained in detail.
- In line 203, is there a clerical error in “Agilent 3670A”?
- Is formula (8) incorrect? At first, I assumed that the expression was written correctly, but the two sides clearly have different dimensions. This discrepancy needs to be explained.
- Does the definition of AEF come from any specific source? If so, the authors should cite the source here. If not, please explain the situation.
- In Figure 7, the authors should arrange the voltage levels either from small to large or from large to small. This would be more formal and easier to follow.
- In line 344, the authors write “AV=55.” The unit “dB” should be added after 55.
- In Table 4, the authors aim to highlight the advantages of the amplifier's compact area. However, reference 39 provides the area of all acquisition chains, making the comparison seem unfair. The authors need to either indicate this discrepancy or replace the comparison items.
- AP signals typically occupy a relatively high signal range. The high-pass frequency measured in this paper is 150Hz, and no tuning measures are provided. In this case, LFP signals may be obtained. Could this affect the acquisition system?
- Generally, it is appropriate to collect AP signals with integral noise less than 6. However, the integral noise in this paper is a bit too high. Will this have any impact?

Author Response
1- In line 88, the authors mention “Electrode Offset Tolerance (EOT) and input impedance.” However, the following section only characterizes the input impedance, and I do not see an explanation of the EOT.
Thank you for your valuable comment. There was an issue with the title, which has now been corrected. The text has been revised to clearly define Electrode Offset Tolerance (EOT) and to separately emphasize the importance of input impedance measurement in Section 2.
Electrode Offset Tolerance (EOT): The amplifier must eliminate DC-offset voltages as high as 50~mV that may emerge at the recording system's input due to the electrochemical nature of the electrode-tissue interface~\cite{chandrakumar201780}. Eliminating this offset is crucial to avoid amplifier circuit saturation and ensure precise signal processing.
2- In lines 99-102, the authors write “First…, Second…, Third….” The first and third points may describe the same amplifier structure, as exemplified by the work of Professor Harrison, which the authors subsequently reference. If the authors intend to describe three different structural features, either the first or third point needs to be explained in detail.
We sincerely appreciate the reviewer's valuable feedback regarding the AC coupling implementation. The three points mentioned are components of the AC coupling technique, which are explained in detail in (DOI: 10.1109/JPROC.2008.922581)
To improve clarity, we have revised the text as follows:
A common approach to achieving low-frequency suppression in amplifier design is the use of capacitive feedback networks or AC coupling. This technique places large capacitors in series with the input electrode to block DC offsets while maintaining signal integrity. Additionally, RC high-pass filtering is employed, utilizing the electrode-electrolyte capacitance in conjunction with a large resistor connected between the amplifier input and ground to reject low-frequency components effectively. This technique, used notably in the
highly cited works of Prof. R. Harrison and Prof. Roman Genov, is known for their simple architectures, offering excellent noise performance with minimal power consumption due
to their single-stage preamplifier designs [ 1, 20– 24 ].
3- In line 203, is there a clerical error in “Agilent 3670A”?
Thank you for pointing this out. The correct model number is 35670A, and we have updated the text accordingly.
4- Is formula (8) incorrect? At first, I assumed that the expression was written correctly, but the two sides clearly have different dimensions. This discrepancy needs to be explained.
We appreciate your insightful comment. In this formula, VDSA represents the reverse gain monitored by the DSA, which was not explicitly stated in the text when VSource was fixed at 5V. We acknowledge the mistyped formula, and it has now been corrected and properly explained in the revised manuscript.
5- Does the definition of AEF come from any specific source? If so, the authors should cite the source here. If not, please explain the situation.
The definition of the Area Efficiency Factor (AEF) is not derived from a specific prior source but is introduced in this work as a new figure-of-merit to evaluate the trade-off between silicon area and amplifier performance. While existing metrics such as NEF (Noise Efficiency Factor) and PEF (Power Efficiency Factor) are commonly used to assess power and noise efficiency, AEF provides a complementary perspective by incorporating area constraints, which are critical for high-density neural recording systems.
This explanation is included in section 5.8
6- In Figure 7, the authors should arrange the voltage levels either from small to large or from large to small. This would be more formal and easier to follow.
Thanks! It has been modified.
7- In line 344, the authors write “AV=55.” The unit “dB” should be added after 55.
Thanks! It has been modified.
8- In Table 4, the authors aim to highlight the advantages of the amplifier's compact area. However, reference 39 provides the area of all acquisition chains, making the comparison seem unfair. The authors need to either indicate this discrepancy or replace the comparison items.
Thank you for your valuable comment. We acknowledge the discrepancy you pointed out regarding the comparison in Table 4. The reference (Muller, R., Gambini, S., & Rabaey, J. M. (2011)) presents a DC-coupled amplifier that mitigates the offset using a mixed signal feedback loop (Fig. 4). Each amplifier in that reference is paired with two DACs to cancel DC offset and LFP signals, and the digital filter is implemented using an off-chip FPGA (Fig. 15). Therefore, the area specified in their work reflects the entire acquisition chain, including DACs and the digital filter.
So, in line with Table 4 of the reference, a complete biopotential amplifier in their work, including the merged amplifier-DAC, summing amplifier&DAC, and digital filter, results in an estimated area about 0.013 mm².
9- AP signals typically occupy a relatively high signal range. The high-pass frequency measured in this paper is 150Hz, and no tuning measures are provided. In this case, LFP signals may be obtained. Could this affect the acquisition system?
We thank the reviewer for raising this important point about potential LFP interference in our system. Below, we clarify our design choices and validate the system’s performance under these constraints:
Our system includes a spike comparison unit, which ensures that the recorded action potential (AP) signals are effectively distinguished from local field potentials (LFPs). While the high-pass corner frequency is set at 150 Hz without additional tuning mechanisms, this cutoff still effectively attenuates the majority of LFP components, which predominantly occupy frequencies below 100 Hz. Any residual LFP content that might be present in the recorded signal is further mitigated by the spike detection and processing stages.
Moreover, AP signals generally exhibit much higher amplitudes and sharper temporal features than LFPs, making them easily separable within the acquisition system. Future implementations may explore tunable high-pass filtering to provide additional flexibility in different recording conditions. However, the current design ensures that AP signals remain the primary focus of neural recording, minimizing any significant interference from LFPs.
The last paragraph of section 6.1 is modified as follows:
The difference between the simulated and measured \emph{f\textsubscript{HP-3dB}} is notable, as illustrated in Figure~\ref{DC_canceller} (a). This discrepancy is likely due to the considerable impact of variations in \(R_{eq}\), as well as the gain of OTA\textsubscript{2} and the capacitance charge from the measurement probe, which can shift \emph{f\textsubscript{HP-3dB}} to a lower frequency. However, this cutoff still effectively attenuates the majority of LFP components, which predominantly occupy frequencies below 100~Hz. Any residual LFP content that might be present in the recorded signal is further mitigated by the spike detection and processing stages.
10- Generally, it is appropriate to collect AP signals with integral noise less than 6. However, the integral noise in this paper is a bit too high. Will this have any impact?
Thank you for your valuable comment on this case. We tried to explain in detail and discuss further in section 7.2 as follows:
The proposed neural amplifier achieves an input-referred noise of 11.1 µV\textsubscript{rms}, which, while slightly higher than some referenced designs (Table 4), remains sufficient for detecting spikes with amplitudes exceeding 50 µV. As demonstrated by Buzsáki et al. (2012), extracellular spike amplitudes typically range from ~50 µV to 500 µV, with most spikes from nearby neurons (>50 µm) surpassing 70 µV. In high-density recording systems (e.g., Utah arrays, Neuropixels), where electrodes are positioned close to active neural populations, the majority of detectable spikes are expected to exhibit amplitudes well above this threshold. Thus, the amplifier’s noise performance remains compatible with reliable spike detection in such configurations.
A key advancement of this work is the significant reduction in silicon area compared to prior designs. For instance, the proposed architecture reduces chip area by at least fivefold relative to Müller et al. (2011), enabling a 2.2× decrease in inter-electrode spacing. This miniaturization facilitates higher electrode density, which offers two critical benefits:
- Improved recording spatial resolution: Closer electrode spacing enhances the ability to resolve individual neural units and reduces signal cross-talk.
- Enhanced signal-to-noise ratio (SNR): By positioning electrodes nearer to neural sources, the amplitude of recorded spikes increases, improving detectability and selectivity.
Future work will involve in vivo validation to assess the amplifier’s performance in biological settings, particularly its ability to capture high-fidelity neural signals in densely packed electrode arrays. These experiments will further clarify the trade-offs between noise, area efficiency, and spike detection efficacy in next-generation neural recording systems.

Reviewer 2 Report
Comments and Suggestions for Authors
Comments to the Author
This paper designs and implements a low-power biopotential amplifier using TSMC 28nm CMOS process with ultra-high input impedance and compact chip area, and optimizes power consumption and noise performance to make it suitable for high-density neural signal recording systems. Some minor comments would like to provide here for authors' reference.
- Formula (8) is used to calculate the input impedance, but does not explain how the VDSA (voltage measurement) is measured in the experiment and whether the calculation method is affected by parasitic capacitance? Please explain.
- The influence of temperature on the Gain is mentioned in "Frequency Bandwidth and Mid-Band Gain" (FIG. 8), but its influence on key parameters such as noise and input impedance is not discussed. It is suggested to supplement the analysis of the influence of temperature change on the overall circuit performance.
- Low-power biological amplifiers are usually used for long-term implantation, and only transient tests are carried out in this paper. It is suggested to supplement the analysis of power stability and gain drift under long-term operation to improve the practicability of the circuit.
- In "Comparison with Previous Studies" (Table 4), the comparison of NEF, PEF, AEF and other key indicators is not visualized, but only listed in numerical form, which is difficult to make an intuitive comparison. It is recommended to add a bar chart or radar chart to more clearly show the advantages and disadvantages of different designs.
- The background of low-power neural signal amplifiers is discussed in the introduction section, but the application of memristors is not mentioned. In recent years, memristors have made remarkable progress in biomedical signal processing due to their synaptic plasticity and low power consumption. It is recommended to supplement relevant research and cite literature to enhance the integrity of the background.Some specific results include:
DOI: 10.1109/TNNLS.2025.3539842
DOI: 10.1109/TCSI.2025.3547336
DOI: 10.1109/TNNLS.2023.3348553
DOI: 10.1109/TASE.2023.3326461

Author Response
1
Formula (8) is used to calculate the input impedance, but does not explain how the VDSA (voltage measurement) is measured in the experiment and whether the calculation method is affected by parasitic capacitance? Please explain.
We appreciate your insightful comment on this issue. Equation (8) has been modified as follows:
where is the gain measured by the DSA during the experiment actually the Vsource is set at 5 V and DSA measure VInput by sweeping a range of frequency.
The parasitic effect for measuring input impedance is also discussed in section 6.5 as follows:
The comparison between measured and simulation input impedance shows that PCB routing, PAD, and ESD parasitic configures the parallel connection with the input impedance of the analog front-end and has the potential to reduce the total input impedance.
2
The influence of temperature on the Gain is mentioned in "Frequency Bandwidth and Mid-Band Gain" (FIG. 8), but its influence on key parameters such as noise and input impedance is not discussed. It is suggested to supplement the analysis of the influence of temperature change on the overall circuit performance.
Thanks for your comment. More detailed diagrams are added to Figure. 8 to shows the sensibility of input referred noise, power consumption, and input referend noise to the temperature variation.
3
Low-power biological amplifiers are usually used for long-term implantation, and only transient tests are carried out in this paper. It is suggested to supplement the analysis of power stability and gain drift under long-term operation to improve the practicability of the circuit.
We appreciate the reviewer’s suggestion regarding the analysis of power stability and gain drift under long-term operation. To address this concern, we conducted additional measurements over a 24-hour period while maintaining a constant temperature. The results indicate no significant variation in the noise level or midband gain, demonstrating the stability of power consumption and amplifier performance over extended operation. These findings confirm the practicality of our circuit for long-term implantation applications.
4
In "Comparison with Previous Studies" (Table 4), the comparison of NEF, PEF, AEF and other key indicators is not visualized, but only listed in numerical form, which is difficult to make an intuitive comparison. It is recommended to add a bar chart or radar chart to more clearly show the advantages and disadvantages of different designs.
Thank you for your valuable comment. We appreciate your suggestion, and as a result, Figure 13 has been added to the manuscript.
5
The background of low-power neural signal amplifiers is discussed in the introduction section, but the application of memristors is not mentioned. In recent years, memristors have made remarkable progress in biomedical signal processing due to their synaptic plasticity and low power consumption. It is recommended to supplement relevant research and cite literature to enhance the integrity of the background.Some specific results include:
DOI: 10.1109/TNNLS.2025.3539842
DOI: 10.1109/TCSI.2025.3547336
DOI: 10.1109/TNNLS.2023.3348553
DOI: 10.1109/TASE.2023.3326461
We appreciate the reviewer’s insightful comment on memristors. As we have recognized, memristors offer significant advantages for AC coupling techniques, which aligns with our future work on developing a modified version of a low-noise, ultra-compact biopotential amplifier.
Memristors enable adaptive offset cancellation by serving as tunable resistive elements in AC coupling or capacitive feedback networks. Their compact size and programmable resistance allow for the replacement of bulky passive components, contributing to lower power dissipation and reduced die area. Furthermore, their synaptic-like behavior facilitates bio-inspired signal processing, enhancing real-time filtering and adaptive gain control for improved neural data acquisition.
The references suggested by the reviewer (DOI: 10.1109/TNNLS.2023.3348553, DOI: 10.1109/TNNLS.2025.3539842) provide valuable insights and will serve as informative support for our work. We sincerely appreciate the recommendation.
Round 2
Reviewer 2 Report
Comments and Suggestions for Authors
The authors have answered all my queries, and in my personal opinion it can be published in this form.
Comments on the Quality of English LanguageThe authors have answered all my queries, and in my personal opinion it can be published in this form.